# The Inclusion of Concentrate with Quebracho Is Advisable in Two Forage-Based Diets of Ewes According to the In Vitro Fermentation Parameters

**DOI:** 10.3390/ani9070451

**Published:** 2019-07-17

**Authors:** Pablo Jose Rufino-Moya, Mireia Blanco, Sandra Lobón, Juan Ramon Bertolín, Ramón Armengol, Margalida Joy

**Affiliations:** 1Instituto Agroalimentario de Aragón—IA2, Centro de Investigación y Tecnología Agroalimentaria de Aragón (CITA), CITA-Universidad de Zaragoza, Avda. Montañana 930, 50059 Zaragoza, Spain; 2Departament de Ciència Animal, UdL-Agrotecnio Center, Avda. de l’Alcalde Rovira Roure 191, 25198 Lleida, Spain

**Keywords:** condensed tannins, gas production, methane production, fresh forage, hay, in vitro, organic matter degradability

## Abstract

**Simple Summary:**

In dry, mountainous areas, ewes are fed low-quality forages (hay or straw) indoors, although they also graze in high-quality pastures when available. Concentrate supplementation is recommended to cover high nutritional requirements during lactation. Condensed tannins (CT) of quebracho (*Schinopsis balansae*) can be included in the concentrate to reduce methane (CH_4_) emissions and ruminal degradation of protein, improving the fermentation’s efficiency. Furthermore, low levels of quebracho can improve some meat and milk quality traits. The effects of the inclusion of concentrate and CT in diets depend on the level of inclusion and the quality of forage. The aim of this study was to evaluate the in vitro fermentation characteristics of diets in order to identify the most suitable one to be fed to ewes under farming conditions. The diets examined differed in quality of the forage available, comparing hay versus fresh forage diets and forage alone versus 70:30 forage:concentrate with (quebracho) or without CT (control) in each type of forage. The fresh-forage-based diets had lower gas and CH_4_ production and greater in vitro organic matter degradability (IVOMD) than the hay-based diets. The inclusion of quebracho concentrate increased the IVOMD in hay-based diets and reduced ammonia content in fresh-forage-based diets.

**Abstract:**

Ewes receive hay or graze on fresh pastures supplemented with concentrates to fulfil their lactation requirements. Quebracho (*Schinopsis balansae*) can be added to change the ruminal fermentation. Fermentation parameters of forages alone and 70:30 forage:concentrate diets with control and quebracho concentrate were compared after 24 h of in vitro incubation. Fresh forage diets produced less gas (*p* < 0.05) and had greater IVOMD (*p* < 0.001), ammonia (NH_3_-N) content, valeric acid, branched-chain volatile fatty acid proportions, and lower propionic acid proportion than the hay diets (*p* < 0.01). In the hay diets, methane production increased with control concentrate (*p* < 0.01) and tended to decrease with quebacho concentrate (*p* < 0.10). The inclusion of both concentrates increased the acetic:propionic ratio (*p* < 0.01), and only the inclusion of quebracho concentrate increased the IVOMD (*p* < 0.01). In the fresh forage diets, gas and methane production increased with the inclusion of the control concentrate (*p* < 0.05), but methane production decreased with quebracho concentrate (*p* < 0.01). The inclusion of quebracho concentrate reduced the NH_3_-N content and valeric acid proportion (*p* < 0.05). In conclusion, the inclusion of quebracho concentrate would be advisable to reduce the CH_4_ production and NH_3_-N content in fresh forage diets and to increase the IVOMD in hay diets in comparison with the forages alone.

## 1. Introduction

In dry, mountainous areas, ewes are usually stalled around parturition and fed cheap, low-quality forages (hay or straw) supplemented with concentrates/grain throughout lactation to fulfil their maintenance and lactation requirements for energy. However, ewes can raise their suckling lambs at pastures during spring [1,2]. In recent years, the condensed tannins (CT) from quebracho (*Schinopsis balansae*) have been incorporated into the diets of dairy ewes to increase their milk production during lactation and improve the quality of the milk [3]. The inclusion of quebracho in the concentrate of grazing and hay-fed Churra Tensina ewes decreased milk protein content [4], reduced lipid oxidation, and increased α-tocopherol content in the meat of the suckling lambs [5].

The forages fed to ewes in these systems differ greatly in their chemical composition, especially the contents of structural carbohydrates and protein, which increase or reduce the parameters of ruminal fermentation depending on chemical composition [6]. The fermentation of low-quality forages leads to high gas and methane (CH_4_) production [7], to the detriment of microbial biomass synthesis [8]. The inclusion of concentrate in diets to cover the energy requirements during lactation can reduce the production of methane [9] and ammonia in the rumen through the enhancement of the direct incorporation of amino N into microbial proteins [10]. Ruminal fermentation parameters are modified by the presence of CT because they have antimicrobial and antiprotozoal activity, as well as the capacity to bind components of the diet, especially protein [11]. Condensed tannins from quebracho can reduce the production of gas, CH_4_ [12], and NH_3_-N [13], or change the molar proportions of volatile fatty acid (VFA) production [14]. However, the effect of the CT on the parameters of ruminal fermentation depends on the diet, the concentration of CT, and the ruminant species [15,16]. The aim of this study was to compare the fermentation characteristics of diets fed to ewes under farming conditions in order to identify the most suitable ones depending on the quality of the forage available.

## 2. Materials and Methods

### 2.1. Experimental Design

Two forages of different quality (hay vs. fresh forage) and two concentrates with or without quebracho CT (control vs. quebracho) were used in this experiment. The forages alone and the four diets composed of 70:30 forage:concentrate were incubated in vitro for 24 h. The fermentation parameters between both forage-based diets and the effect of the inclusion of control or quebracho concentrate within each forage with respect to the forage alone were compared.

### 2.2. Animal and Diets

#### 2.2.1. Feedstuffs and Substrates

The forage samples were obtained from a meadow located at La Garcipollera Research Station in the mountainous area of the southern Pyrenees (northeastern Spain; 42°37′ N, 0°30′ W; 945 m a.s.l.). The meadow was composed of 20% legumes (mainly *Trifolium repens*), 68% grass (the main species were *Festuca arundinacea*, *Festuca pratensis*, and *Dactylis glomerata*), and 12% other species (mainly *Rumex acetosa* and *Ranunculus bulbosus*). Hay was obtained from forage harvested at the late bud stage (July 2013), as farmers usually do to maximize forage production, and was of a medium–low quality. The forage was sun-dried and baled. Fresh forage samples were collected at the beginning of vegetative growth (March 2014), which is the stage of the forage at the beginning of the grazing period of the ewes in mountainous areas. The isoenergetic (11.7 metabolizable energy MJ/kg DM) and isoproteic (154 g/kg DM crude protein (CP)) concentrates were formulated to fulfil the requirements of lactating ewes. The ingredients of the control concentrate were barley (34%), bran (20%), corn gluten feed (18%), sunflower flour (12%), corn grain (10%), alfalfa pellet (2%), calcium carbonate (2%), molasses (1%), and mineral mix (1%). The ingredients of the quebracho concentrate were barley (49%), corn gluten feed (15%), soybean meal (12%), quebracho extract (10%; SYLVAFEED ByPro Q, Adial Nutrition. Gerona, Spain, with 75% CT), bran (6%), corn grain (5%), calcium carbonate (2%), and mineral mix (1%).

The samples of the feedstuffs were dried in an oven at 60 °C for 48 h to determine the chemical composition, except for a part of the samples that was immediately freeze-dried to determine the CT content and which was to be incubated in the in vitro assays. All the samples were ground (Rotary Mill, ZM200 Retsch, Haan, Germany) and sieved through a 1 mm screen, except for the parts used to determine the CP and CT contents, which were sieved through a 0.2 mm screen. All the samples were stored in total darkness at −20 °C until further analyses.

Both forages and the four diets were in vitro assessed. The feedstuffs used were the abovementioned forages, plus concentrate—control or quebracho concentrate. Therefore, the following substrates were incubated: hay alone (H), fresh forage alone (FF), 70% hay plus 30% control concentrate (HC), 70% hay plus 30% quebracho concentrate (HQ), 70% fresh forage plus 30% control concentrate (FFC), and 70% fresh forage plus 30% quebracho concentrate (FFQ).

#### 2.2.2. Animals and Sampling of Ruminal Digesta

All procedures used in the experiment were carried out in accordance with the Spanish guidelines for experimental animal protection (RD 53/2013) with the approval of the Institutional Animal Care and Use Committee of the Research Centre (procedure number 2011-05). The ruminal contents were collected into a prewarmed insulated thermos before morning feeding from four rumen-cannulated Rasa Aragonesa wethers (65 ± 2.1 kg body weight), fed 70:30 alfalfa hay:barley grain at an energy maintenance level. For more details, read Rufino-Moya et al. [17]. Rumen fluid from the four wethers was mixed, and a buffer solution was added in a proportion of 1:2 (*v*/*v*) based on the protocol of Menke and Steingass [18].

#### 2.2.3. In Vitro Gas Production Technique and Sampling

The Ankom system (Ankom Technology Corporation, Fairport, NY, USA) was used to determine the gas production following the conditions described by Rufino-Moya et al. [17]. Five hundred milligrams of samples were incubated with 60 mL of buffered solution:rumen fluid (2:1, *v*/*v*) in a water bath at 39 °C for 24 h. All bottles were constantly flushed with CO_2_ gas to create an anaerobic environment before the start of incubation, and sealed with the Ankom pressure monitor cap. Three runs were conducted on three days in consecutive weeks. Blanks were included in each run. The samples and blanks were incubated in triplicate in each run. Gas production was recorded for 24 h and corrected with the blanks.

After 24 h of incubation, the bottles were placed in ice to stop fermentation (5–10 min) and then tempered at room temperature for 10–15 min. The sampling to determine CH_4_ production, NH_3_-N content, and VFA proportions was carried out according to Rufino-Moya et al. [17]. Briefly, a sample of gas of headspace at the end of fermentation was collected from each bottle at atmospheric pressure with a syringe attached to a manometer HD2304.0 (Delta Ohm, Padua, Italy) into a Vacutainer^®^ tube (Becton Dickinson, Plymouth, United Kingdom). The tubes were conserved at 4 °C until CH_4_ determination. After gas sampling, the pH of the incubation medium was measured immediately with a microPH 2002 (Crison Instruments S.A., Barcelona, Spain). For NH_3_-N determination, 2.5 mL of liquid was mixed with 2.5 mL HCl 0.1 N, and for VFA determination, 0.5 mL of the liquid was added to 0.5 mL of deproteinizing solution and 1 mL of distilled water. Tubes with gas samples were stored at 4 °C, and those with samples for determination of NH_3_-N and VFA were stored at −20 °C until future analyses. The entire bottle content was filtered through a pre-weighed bag (50 μm; Ankom) to estimate the in vitro organic matter degradability (IVOMD). Briefly, the bags were sealed, washed, dried at 103 °C for 48 h, and finally placed in a muffle at 550 °C to obtain the ashes. The organic matter of the bag content was obtained as dry matter (DM) ashes, and the IVOMD was calculated.

### 2.3. Analytical Methods

#### 2.3.1. Chemical Composition

All the analyses of the chemical composition and secondary compounds were done according to the official methods, as reported in Rufino-Moya et al. [17]. The DM, ash, and CP were determined according to the AOAC methods [19]. Neutral detergent fiber (NDFom), acid detergent fiber (ADFom), and lignin (sa) content were determined using the method of Van Soest et al. [20]. The ether extract (EE) was determined following the Ankom Procedure [21] with an XT10 Ankom extractor (Ankom Technology Corporation, Fairport, NY, USA). The nonstructural carbohydrates (NSC) were calculated as NSC = 1000 − CP − EE − NDF − ash, according to Guglielmelli et al. [22]. Metabolisable energy of the forages was estimated using the equation proposed by Mertens [23] and that of the concentrates according to the manufacturer. The content of total polyphenols (TP) was extracted using the method of Makkar [24] and determined with the method of Julkunen-Tiitto [25]. The extractable CT (ECT), protein-bound CT (PBCT), and fiber-bound CT (FBCT) were extracted according to Terrill et al. [26] and quantified at 550 nm based on the method described by Grabber et al. [27]. The standards used for quantification of the samples were extracted and purified from sainfoin for forages and quebracho (SYLVAFEED ByPro Q) for concentrates using the method described by Wolfe et al. [28].

#### 2.3.2. Determination of the Parameters of In Vitro Fermentation

The gas production recorded hourly for 24 h by the Ankom system was used to estimate the parameters of the kinetics of fermentation, adjusting the gas produced to the model described by France et al. [29]:$$P=A\times \left(1-e^{-ct}\right)$$
where *P* is the cumulative gas production (mL) at time *t* (h), *A* is the potential gas production (mL), and *c* is the rate of gas production (h^−1^).

Methane was determined in an HP-4890 (Agilent, St. Clara, CA, USA) gas chromatograph (GC) equipped with a flame ionization detector (FID) and a TG-BOND Q+ capillary column (30 m × 0.32 mm id × 10 µm film thickness, Thermo Scientific, Waltham, MA, USA). The carrier gas was helium, at a flow rate of 1 mL/min. The temperature of the oven was maintained at 100 °C (isothermal program). The volume of the splitless injection was 200 µL. The identification of CH_4_ was based on the retention time compared with the standard. The concentration was determined from the peak concentration:area ratio using the peak area generated from standard gas as the reference (CH_4_; 99.995% purity [C45], Carburos Metálicos, Barcelona, Spain). The methane production was calculated by the model proposed by Cattani et al. [30] for the Ankom Gas Production System, which has been shown to enable accurate estimation of methane production [31]:$$MP=-0.0064\;\times \;{\left[HSCH_{4}\;\times \;\left(HSV+GP\right)\right]}^{2}+0.9835\;\times \;[HSCH_{4}\;\times \;\left(HSV+GP\right)]$$
where *MP* (mL) is methane production, *HSCH_4_* (L L^−1^) is the methane proportion in the headspace, *HSV* (mL) is the headspace volume, and *GP* (mL) is the total gas production of each bottle.

Ammonia content was determined in an Epoch microplate spectrophotometer (BioTek Instruments, Inc., Winooski, VT, USA) using the colorimetric method described by Chaney and Marbach [32].

The concentration of acetic, propionic, isobutyric, butyric, isovaleric, and valeric acids was measured in a Bruker Scion 460 GC (Bruker, Billerica, MA, USA) equipped with a CP-8400 autosampler, FID, and a BR-SWax capillary column (30 m × 0.25 mm i.d. × 0.25 µm film thickness, Bruker, Billerica, MA, USA). The carrier gas was helium (flow rate of 1 mL/min). The temperature program of the oven was 100 °C, followed by a 6 °C/min increase to 160 °C. The injection volume was 1 µL at a split ratio of 1:50. The identification of the individual VFA was based on retention time comparisons with commercially available standards of acetic, propionic, isobutyric, butyric, isovaleric, valeric, and 4-methyl-valeric acids at ≥99% purity (Sigma-Aldrich, St. Louis, MO, USA).

### 2.4. Calculations and Statistical Analysis

Data were analyzed using the statistical software, SAS V.9.3 (SAS Inst. Inc., Cary, NC, USA). The fermentation kinetic parameters (*A* and *c*) were estimated through a nonlinear regression model using the SAS NLIN program. The total gas, methane, *A*, *c*, IVOMD, and the fermentation end-products were tested by analysis of variance using the general linear model (GLM) procedure of SAS considering the substrate the fixed effect. Orthogonal contrasts were used to compare the difference between type of forage (H, HC, and HQ vs. FF, FFC, and FFQ) and the effect of the inclusion of control or quebracho concentrate on hay-based diets (H vs. HC and H vs. HQ) and fresh-forage-based diets (FF vs. FFC and FF vs. FFQ). The effects were considered significant, or a trend was found at a probability value of *p* < 0.05 and 0.05 ≤ *p* < 0.10, respectively.

## 3. Results

The chemical compositions of the feedstuffs are shown in Table 1. As expected, the hay had higher fiber content and lower CP and NSC content than the fresh forage. The content of condensed tannins in both forages was almost negligible. Regarding the concentrates, both had similar CP but different NDFom, ADFom, lignin (sa), and NSC content. The inclusion of quebracho in the concentrate increased the total polyphenol and CT content.

### 3.1. Effect of the Type of Forage on In Vitro Fermentation

The type of forage (hay vs. fresh forage) had major effects on most of the in vitro fermentation parameters (Table 2). Regarding to the degraded organic matter (dOM), the hay-based diets produced more gas (*p* < 0.05) and CH_4_ (*p* < 0.001) than the fresh-forage-based diets. The type of forage also affected the IVOMD (*p* < 0.001) and the NH_3_-N content (*p* < 0.01), recording lower values in hay-based diets than in fresh-forage-based diets. The total VFA was not affected by type of forage (*p* > 0.05), but the individual VFA proportions varied according to the forage with lower acetic (*p* < 0.10) and propionic acid proportions (*p* = 0.01), but greater proportions of butyric (*p* < 0.01), valeric (*p* < 0.001), and branched chain VFA (BCVFA) (*p* < 0.001) in the fresh-forage-based diets than in the hay-based diets. The type of forage did not affect the C_2_:C_3_ ratio (*p* > 0.05) and the CH_4_/VFA_total_ ratio (*p* > 0.05), with the hay-based diets presenting lower ratios than the fresh-forage-based diets.

### 3.2. Hay-Based Diets: Effect of the Inclusion of the Control or Quebracho Concentrate

The fermentation parameters are shown in Table 2, and the fermentation kinetics during the incubation of the hay-based diets is represented in Figure 1. Gas production (mL/g dOM) and NH_3_-N content were not affected by the inclusion of concentrate, regardless of the presence of quebracho (*p* > 0.05). However, the parameters of the kinetics of fermentation were differently affected depending on the presence of quebracho. The potential gas production decreased only with the inclusion of quebracho concentrate (*p* < 0.05). The rate of gas production *c* tended to increase with the inclusion of the control concentrate (*p* < 0.10), whereas it increased with the inclusion of quebracho concentrate (*p* < 0.01). The CH_4_ production (mL/g dOM) increased with the inclusion of control concentrate (*p* < 0.01) and tended to reduce with quebracho concentrate (*p* < 0.01). Furthermore, the inclusion of concentrate improved the IVOMD to a greater extent with quebracho concentrate (*p* < 0.05), and a tendency was only observed with the inclusion of the control concentrate (*p* < 0.10). The inclusion of concentrates affected most of the molar proportions of VFA (*p* < 0.05), but not total VFA production (*p* > 0.05). The inclusion of both concentrates similarly reduced the proportions of propionic and isovaleric acids (*p* < 0.01) and tended to reduce the proportion of isobutyric acid (*p* < 0.10) but increased butyric acid (*p* < 0.05). The C_2_:C_3_ ratio increased when concentrate was added to a greater extent with the control concentrate (*p* < 0.01) than with the quebracho concentrate (*p* < 0.05).

### 3.3. Fresh-Forage-Based Diet: Effect of the Inclusion of Control or Quebracho Concentrate

The parameters obtained in the in vitro fermentation are shown in Table 2, and the fermentation kinetics during the incubation are represented in Figure 2. The inclusion of the control concentrate increased the production of total gas and CH_4_ (mL/g dOM; *p* < 0.05) without affecting IVOMD and NH_3_-N (*p* > 0.05). However, the inclusion of quebracho concentrate did not affect the production of gas (mL/g dOM; *p* > 0.05) but reduced CH_4_ production (mL/g dOM; *p* < 0.01), and tended to increase IVOMD (*p* < 0.10) and reduce NH_3_-N content (*p* < 0.05). The total VFA and the proportions of acetic and propionic acids were not affected by the inclusion of concentrate (*p* > 0.05). The inclusion of both concentrates tended to increase the proportion of butyric acid (*p* < 0.10) and reduce the proportions of BCVFA (*p* < 0.05 to *p* < 0.001). The proportion of valeric acid was only affected by the inclusion of quebracho concentrate (*p* < 0.05), which decreased its proportion.

## 4. Discussion

### 4.1. Effects of the Type of Forage on Fermentation Parameters

In the present study, the forages for the fresh and hay diets were harvested at different stages of maturity in an attempt to use the forages actually offered to ewes in dry mountainous areas. As stated above, ewes are generally fed hay-based diets indoors when natural resources are scarce, or they graze when the pasture is available. Usually, farmers harvest the forage for haymaking when maximum production is possible with the forage being near maturity, and they turn out the ewes to the meadows at the start of the grazing season, when forages are in the vegetative stage. Therefore, these two forages are either consumed fresh or as hay, and may have different ruminal fermentation characteristics, which may be reflected in different productive performances [4,5].

One of the most important objectives of nutrition is to improve microbial efficiency, thus maximizing feed conversion into microbial biomass, improving microbial protein supply to the small intestine, and, proportionally, reducing energy losses by gas and CH_4_ production [8,33,34]. In the present study, the lower gas and CH_4_ production and greater IVOMD of fresh-forage-based compared with hay-based diets were due to the lower content of structural carbohydrates of the former than the latter [9,35]. These results might indicate that the organic matter degraded in fresh-forage-based diets is mainly destined for microbial biomass synthesis [8,22]. Furthermore, the increase of *c* in fresh-forage-based diets could be related to higher content of rapidly fermentable carbohydrates [35], showing a fast fermentation process which allows for high voluntary intake [36]. Thus, fresh-forage-based diets could increase animal production due to lower energy losses, lower methane production, and greater degradability of the feedstuffs [22,34].

In the current experiment, ammonia content and the proportions of valeric, isobutyric, and isovaleric acids, which were derived from the fermentation of protein and branched-chain amino acids, reflected the CP content of the forages [22,37]. Generally, the presence of structural carbohydrates is associated with the production of acetic and butyric acids to the detriment of propionic acid [6,34], which was not registered in the present study, probably due to the short incubation period in comparison with previous studies. However, as in the current experiment, Saro et al. [37] reported that diets with a higher concentration of fiber had higher proportions of acetic and propionic acids, a lower proportion of butyric acid, and a similar C_2_:C_3_ ratio than diets with a lower concentration of fiber under in vivo conditions until 8 h postfeeding.

### 4.2. Effect of the Inclusion of Concentrate with or without Quebracho

In suckling lamb production systems, concentrates are offered to ewes to cover the high energy requirements of lactation [38] because there is an increase of dietary energy and, thereby, an optimization of the efficiency of feed utilization [39]. Occasionally, CT from quebracho can be included in the concentrate to promote a propionic rumen fermentation towards a lower C_2_:C_3_ ratio and CH_4_ production [12,13,15]. In the current experiment, the concentration of condensed tannins from quebracho was limited to 75 g/kg DM in the concentrate because greater contents lead to a reduction in voluntary feed intake and lesions in digestive tracts [40]. However, the final dose in the 70:30 forage:concentrate diets was 22.5 g/kg DM of CT from quebracho.

The inclusion of concentrate is associated with an increase in gas production [41,42], mainly due to an increase in the amount of energy available to the microbial population during the first few hours of incubation [43]. In the current experiment, this effect was noticeable in the first few hours of incubation of hay-based diets (Figure 1). However, the effect on gas production (mL/g dOM) at 24 h was only evident with the inclusion of the control concentrate in the fresh forage diet. The absence of the effect of the inclusion of concentrate at the end of incubation in hay-based diets is in agreement with Zicarelli et al. [44], who observed similar gas production with the inclusion of 30% concentrate in a diet with oat hay. The inclusion of quebracho concentrate had no effect on gas production with respect to forage alone, as reported with the addition of 30 g/kg DM of quebracho extract in a silage-based diet (67:33 forage:concentrate) [45]. Similarly, the inclusion of 20 g/kg DM CT from quebracho in a total mixed ration had no effect on gas production [15].

The CH_4_ production increased with the inclusion of control concentrate in both forage-based diets, agreeing with Pedreira et al. [46], who recorded an increase of CH_4_ production with sorghum silage supplemented with 30% of concentrate. However, the reduced CH_4_ production with the inclusion of quebracho concentrate in both types of forage-based diets could be due to the presence of CT that counterbalanced the presence of rapidly fermentable carbohydrates in the concentrate, which accelerated the degradation process and increased bacterial growth [47,48] during the short incubation in the present study. In contrast, Hassanat and Benchaar [15] did not report any effect on CH_4_ production when a mixed ration with 20 g/kg DM CT from quebracho was incubated in cow ruminal fluid. This discrepancy might be due to the different diet and donor animal of ruminal fluid [16].

In the current experiment, the inclusion of concentrates increased the IVOMD in the hay-based diet, as has been reported for low-quality forages [42,44]. Surprisingly, the inclusion of quebracho concentrate increased the IVOMD, whereas a higher dose of quebracho CT extract (30 g/kg DM) reduced the IVOMD of alfalfa hay-based diets [45] due to the linkage with protein or polysaccharide molecules in the feed [11]. Thus, the high content of nonstructural carbohydrates of the quebracho concentrate could have contributed to the improvement in the IVOMD.

In the present experiment, the inclusion of concentrates did not affect NH_3_-N content in the hay-based diets with low CP content, which is in agreement with other authors [13,15]. The effect of the inclusion of concentrate on NH_3_-N production varied according to the presence of quebracho in the fresh forage diet. The similar NH_3_-N concentration of the control concentrate when compared to the forage alone agrees with Archimède et al. [49], who included 30% and 60% of concentrate in alfalfa hay-based diets with a high protein content. However, the inclusion of quebracho concentrate reduced the NH_3_-N content, as reported by Castro-Montoya et al. [45], who observed the reduction of NH_3_-N content with the inclusion of 15 g/kg DM of quebracho. This reduction could be associated with a reduction of protein degradation due to the presence of CT [11].

Results showed that neither the inclusion of concentrate nor the addition of quebracho affected the total VFA production, according to previous studies that used similar proportions of the concentrate [44,46] and a similar inclusion of quebracho [13,15]. Most works in the literature report no effect or a reduction of C_2_:C_3_ due to the increase of the propionic proportion [15,44,46,49], with greater inclusion of concentrate and quebracho than the ones used in the present experiment. The low inclusion of CT from quebracho in the current experiment did not reduce the C_2_:C_3_ ratio, as it was reported in previous studies [15,45]. The reduction of the proportion of valeric acid with the inclusion of quebracho concentrate in the forage-based diet and the BCVFA with the inclusion of concentrates in both forages agrees with the enhancement of the microbial capture of these VFAs due to the energetic input from the concentrates [41,50]. Furthermore, the presence of CT can increase the utilization of BCVFA for microbial protein synthesis [14] and/or can reduce protein degradation in the rumen [13,15,45].

## 5. Conclusions

In conclusion, fresh-forage-based diets had lower gas and CH_4_ production in relation to degraded organic matter, and higher IVOMD than hay-based diets. The inclusion of control concentrate increased CH_4_ production in both forage-based diets and gas production—though only in fresh-forage-based diets—without affecting IVOMD. It would be advisable to include CT from quebracho at a low dose (22.5 g CT/kg DM in diets) because it reduced the CH_4_ production and the protein degradation in fresh forage-based diets and increased the IVOMD and tended to reduce the CH_4_ production in hay-based diets.

## Figures and Tables

**Figure 1 animals-09-00451-f001:**
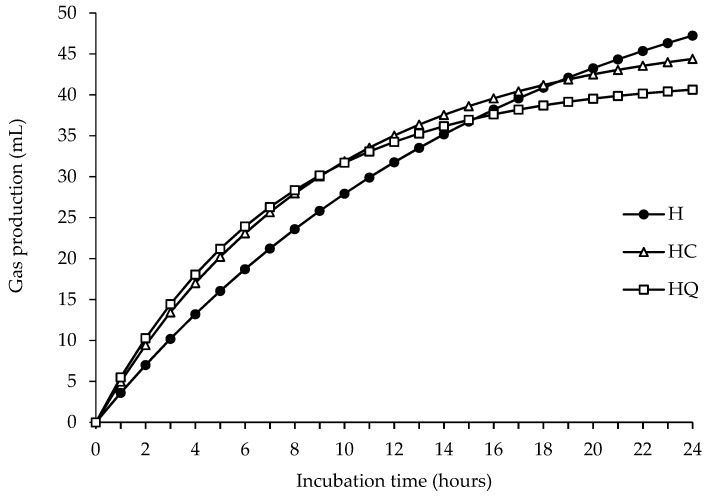
Fermentation kinetics of the hay-based diets during 24 h of incubation. H: 100% hay; HC: 70% hay:30% control concentrate; HQ: 70% hay:30% quebracho concéntrate.

**Figure 2 animals-09-00451-f002:**
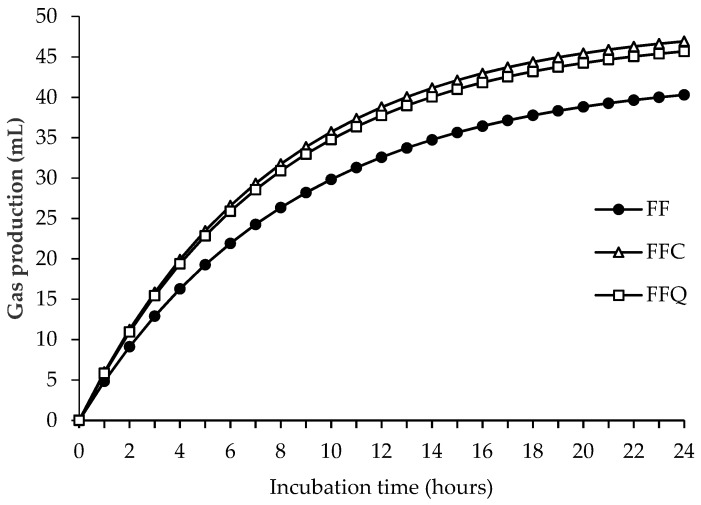
Fermentation kinetics of the fresh-forage-based diets during 24 h of incubation. FF: 100% fresh forage; FFC: 70% fresh forage:30% control concentrate; FFQ: 70% fresh forage:30% quebracho concentrate.

**Table 1 animals-09-00451-t001:** Chemical composition of feedstuffs.

Parameters	Forage		Concentrate	
Items	Hay	Fresh	Control	Quebracho ^1^
Dry matter (DM) (g/kg)	883	183	887	885
Ash (g/kg DM)	83	13	72	71
CP (g/kg DM)	75	263	153	155
NDFom (g/kg DM)	678	488	270	193
ADFom (g/kg DM)	362	202	85	66
Lignin (sa) (g/kg DM)	43	40	22	12
Ether extract (g/kg DM)	18	33	31	30
Nonstructural carbohydrates (g/kg DM)	146	203	474	551
Metabolizable energy ^2,3^	8.6	11.9	11.7	11.6
Total polyphenols (eq-g tannic acid/kg DM)	7.3	12.7	7.2	66
Condensed tannins (CT) (eq-g CT/kg DM) ^4^				
Total CT	1.5	1.8	10.5	76.6
Extractable CT	0.4	0.9	7.3	72.2
Protein-bound CT	0.7	0.6	2	2.5
Fiber-bound CT	0.3	0.3	1.1	2

^1^ Concentrate with 10% quebracho; ^2^ Estimated according to the equation proposed by Mertens [23] in forages; ^3^ calculated according to data of the manufacturer; ^4^ in forages expressed as sainfoin CT equivalents; in the concentrates expressed as quebracho CT equivalents.

**Table 2 animals-09-00451-t002:** Effect of the type of forage and the inclusion of control or quebracho concentrate on gas and CH_4_ production, in vitro organic matter digestibility (IVOMD), NH_3_-N, and volatile fatty acids (VFA) after 24 h of incubation.

Parameters	Substrates ^1^							(*p*-Value)		Contrast (*p*-Value)			
	H	FF	HC	HQ	FFC	FFQ	RMSE ^2^	Substrates	Hs vs. FFs ^3^	H vs. HC	H vs. HQ	FF vs. FFC	FF vs. FFQ
pH	6.65	6.61	6.67	6.69	6.69	6.77	0.14	0.82	0.81	0.88	0.73	0.51	0.19
Total gas production (mL/g iDM ^4^)	83	88	99	93	107	102	7.3	0.01	0.04	0.023	0.13	0.007	0.04
Total gas production (mL/g dOM ^5^)	181	150	191	172	175	163	14	0.04	0.013	0.42	0.43	0.04	0.27
Potential gas production (*A*) (mL)	61.9	42.7	47.8	42.1	49.1	47.8	10.4	0.28	0.42	0.12	0.04	0.47	0.56
Rate of gas production (*c*) (h^−1^)	0.06	0.12	0.11	0.14	0.13	0.13	0.03	0.04	0.06	0.06	0.005	0.84	0.82
Total CH_4_ production (mL/g iDM)	44	48	49	48	51	49	1.4	0.001	0.012	<0.001	0.004	0.02	0.42
Total CH_4_ production (mL/g dOM)	92.0	81.6	95.9	89.1	82.8	77.6	4.9	0.005	<0.001	0.004	0.087	0.04	0.004
CH_4_/gas (mL/L)	300	321	310	311	297	298	12.9	0.25	0.807	0.38	0.36	0.04	0.06
IVOMD (g/kg)	498	668	554	582	703	723	37.7	<0.001	<0.001	0.097	0.02	0.28	0.099
NH_3_-N (mg/L)	105	134	87	91	118	103	16	0.03	0.007	0.2	0.31	0.23	0.03
Total VFA (mmol/L)	58.5	62.7	63.5	61	61.3	59	5.49	0.85	0.99	0.29	0.59	0.75	0.42
Acetic acid (C_2_) (mmol/mol)	684	678	697	693	682	679	11.34	0.31	0.06	0.18	0.33	0.7	0.94
Propionic acid (C_3_) (mmol/mol)	159	142	142	145	138	143	5.23	0.007	0.01	0.002	0.008	0.41	0.85
Butyric acid (mmol/mol)	98	109	109	111	118	119	5.93	0.014	0.007	0.04	0.03	0.098	0.06
Valeric acid (mmol/mol)	14.9	17.6	14.1	13.5	16.7	15.4	1.05	0.004	<0.001	0.35	0.11	0.33	0.02
Isobutyric acid (mmol/mol)	15.4	18	13.7	13.8	16	15.8	0.98	0.002	<0.001	0.06	0.06	0.03	0.02
Isovaleric acid (mmol/mol)	28.9	35.2	24.1	23.9	29.4	28	1.94	<0.001	<0.001	0.011	0.009	0.003	0.001
C_2_:C_3_ ratio (mol/mol)	4.33	4.79	4.94	4.8	4.93	4.76	0.23	0.059	0.22	0.006	0.03	0.47	0.86
CH_4_/VFA_total_ (mL/mmol)	5.90	5.87	6.02	6.08	6.44	6.40	0.52	0.64	0.36	0.78	0.68	0.20	0.23

^1^ H: 100% hay; FF: 100% fresh forage; HC: 70% hay:30% control concentrate; HQ: 70% hay:30% quebracho concentrate; FFC: 70% fresh forage:30% control concentrate; FFQ: 70% fresh forage:30% quebracho concentrate; ^2^ Root mean standard error; ^3^ Hs vs. FFs: H, HC, and HQ vs. FF, FFC, and FFQ; ^4^ incubated dry matter; ^5^ degrade organic matter.

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
