# Peer review of "The Inclusion of Concentrate with Quebracho Is Advisable in Two Forage-Based Diets of Ewes According to the In Vitro Fermentation Parameters"

_animals, 2019, doi:10.3390/ani9070451_

Round 1
Reviewer 1 Report
Row 114: after 30 control… add the word "concentrate";
Row 218 table 1: Lignin value of fresh forage is very high, please check it.
Row 245: change the word "there" with "the";
Row 261 table 2: After RMSE column add a new column with P probably of "Substrate" factor.
Rows 310-312: Correctly the authors say "Generally, the presence of structural
carbohydrates is associated with the production of acetic and butyric acids to the detriment of
propionic acid [6,31], which did not occur in the present study". In my opinion Authors should give a possible motivation to this particular result.
Author Response
The answers to comments from reviewer 1 has been uploaded

Reviewer 2 Report
I have made comments on the attached version of the manuscript. The manuscript was not easy to review because line numbers did not appear on all pages. I think the English language could be considerably improved. The authors should avoid making statements that are unclear or "empty". They should not say "something affected something". They should say: "something increased or decreased something. The methods section needs attention. Were the bottles flushed with N2 gas? How was the gas sample collected? How was methane production estimated? This should be recalculated as shown by Hannah et al or Alvarez-Hess et al. The conclusion should be based on results and undefined words like "efficiency" should be avoided.

Author Response
The answers to comments from reviewer 2 has been uploaded

Round 2
Reviewer 2 Report
Animals 521925 Revised manuscript June 2019 by Rufino-Moya et al.
The authors have made the majority of edits suggested by this reviewer and the manuscript is much improved.
There are still a small number of minor issues, mainly about English expression that should be improved.
:
Line 193: …. Ankom Gas Production System, which has been shown to enable accurate estimation of methane production [31]:
Line 226: The chemical compositions of the feedstuffs are shown in Table 1.
Table 3: CH4/gas (ml/L) Note there seems to be some errors of calculation for CH4/gas (ml/L). For example, for the H diet, the authors show 300 ml CH4/L gas, but they also report 44 ml of methane production and 83 ml of total gas production per gram of DM incubated. Thus, this indicates there was 1000* 44/83 = 530 ml/L . This number is much larger than the 300 ml/L reported in the table. Also, all of the other reported numbers for this parameter seem too low based on this calculation. It is suggested the authors check their calculations for this parameter and report correct numbers. Also, it should be noted in the text that in this experiment, methane concentration in fermentation gas in excess of 500 ml/L were estimated and this finding is unusual (see Hannah et al. 2016 Anim Prod Sci 56, 244-251). The authors should discuss possible reasons for this finding.
Also, please check the calculation of CH4/VFAtotal (mL/mmol) shown at the bottom of Table 2.
Line 285: … the incubation are represented in Figure 2.
Line 322: fresh forage-based diets could increase animal production because ….
Line 343: reduction in voluntary feed intake …
Line 358: …. Control concentrate in both forage-based diets
Line 364: … In contrast, Hassanat and Benchaar [15] did not report an effect on methane production when a mixed ration with 20 g/kg DM CT from quebracho was incubated in ruminal fluid.
Line 373: molecules in the feed
Author Response

(The authors gave the same response as above.)
